# Eupatorin and Salviandulin-A, with Antimicrobial and Anti-Inflammatory Effects from *Salvia lavanduloides* Kunth Leaves

**DOI:** 10.3390/plants11131739

**Published:** 2022-06-30

**Authors:** Manasés González-Cortazar, David Osvaldo Salinas-Sánchez, Maribel Herrera-Ruiz, Dionisio Celerino Román-Ramos, Alejandro Zamilpa, Enrique Jiménez-Ferrer, Ever A. Ble-González, Patricia Álvarez-Fitz, Ricardo Castrejón-Salgado, Ma. Dolores Pérez-García

**Affiliations:** 1Centro de Investigación Biomédica del Sur, Instituto Mexicano del Seguro Social, Argentina No. 1, Col. Centro, Xochitepec 62790, Morelos, Mexico; gmanases2000@gmail.com (M.G.-C.); edanae10@yahoo.com.mx (M.H.-R.); azamilpa_2000@yahoo.com.mx (A.Z.); enriqueferrer_mx@yahoo.com (E.J.-F.); 2Centro de Investigación en Biodiversidad y Conservación (CIByC), Universidad Autónoma del Estado de Morelos (UAEM), Av. Universidad 1001, Col. Chamilpa, Cuernavaca 62209, Morelos, Mexico; 3Escuela de Estudios Superiores del Jicarero (EESJ), Universidad Autónoma del Estado de Morelos, Carretera Galeana-Tequesquitengo s/n Col. el Jicarero, Jojutla 62909, Morelos, Mexico; roman.ramos.dionisio@gmail.com; 4División Académica de Ciencias Básicas, Universidad Juárez Autónoma de Tabasco, Carretera Cunduacán-Jalpa Km. 0.5, Cunduacán 86690, Tabasco, Mexico; ble_49@hotmail.com; 5Laboratorio de Toxicologia, Cátedra CONACyT-Universidad Autónoma de Guerrero, Av. Lázaro Cárdenas s/n. Col. La Haciendita, Chilpancingo 39070, Guerrero, Mexico; paty_fitz@hotmail.com; 6Instituto Mexicano del Seguro Social, Unidad de Medicina Familiar Número 3, Avenida Insurgentes Esquina con Emiliano Zapata s/n., Centro, Jiutepec 62550, Morelos, Mexico; carisal13@hotmail.com

**Keywords:** *Salvia lavanduloides*, minimum inhibitory concentration (MIC), 13-acetate of 12-O-tetradecanoylforbol (TPA)

## Abstract

This study describes the antimicrobial and anti-inflammatory effects from extracts obtained from the leaves of *Salvia lavanduloides*. The plant material was macerated with three solvents of ascending polarity (*n*-hexane (Sl-Hex), ethyl acetate (Sl-AcOEt), and dichloromethane (Sl-D)). The extracts, fractions (SlD-2 and SlD-3), and isolated compounds (15,16-epoxy-10-β-hydroxy-neo-cleroda-3,7,13(16),14-tetraene-17,12R:18,19-diolide (**1**), salviandulin A (**2**), and eupatorin (**3**)) were evaluated as antimicrobials against Gram-negative, Gram-positive bacteria and the fungus *Candida albicans* (Ca) using the minimum inhibitory concentration (MIC) and the anti-inflammatory activity induced by 13-acetate of 12-O-tetradecanoylforbol (TPA). Sl-D and Sl-AcOEt extracts, SlD-2 and SlD-3 fractions showed the highest antimicrobial activity. The isolated compounds showed good activity against *Pseudomonas aeruginosa* with a MIC < 2 μg/mL, while the anti-inflammatory activity, the Sl-Hex, Sl-D extracts, and SlD-3 fraction presented an inhibition of 62, 45 and 61%, respectively, while (**2**) 70% and (**3**) 72%.

## 1. Introduction

One of the largest problems faced by the public health sector is infections transmitted within hospitals, *Staphylococcus aureus* being the main cause of infections after surgery, followed by pneumonia and intrahospital bacteremia, causing up to 50% mortality [1]. In Mexico, healthcare-related infections caused 32 deaths per 100,000 inhabitants, and it is estimated that prevalence can reach up to 23% in intensive care units [2]. It is generally known that many microorganisms are achieving antibiotic resistance, including *Staphylococcus aureus* and *Pseudomonas aeruginosa*. Therefore, the WHO has created a list of pathogens that need new pharmaceutical drugs to mitigate this problem [3]. *S. aureus* is an opportunistic pathogen, and it is one of the most dangerous staphylococcal bacteria. This Gram-positive bacterium can be part of the human microflora; it has been found on the skin and nasopharynx, affecting immunocompromised patients, causing skin, nose, urethra, vagina, and gastrointestinal tract infections. If mistreated, those infections can lead to pneumonias, endocarditis, and bone infections [4]. *P. aeruginosa* is also an opportunistic bacterium, easily infecting vulnerable patients with severe medical problems, such as otitis externa, ecthyma gangrenous, folliculitis, soft tissue infection, among others [5]. The moment a tissue is affected by any pathogenic bacteria, the response of cytokines, chemokines, lipid mediators, and bioactive amines is activated by various types of cells, such as monocytes, dendritic cells, natural killer cells, and mast cells, activating the release of pro-inflammatory molecules [6]. Inflammation is a process involved in the pathogenesis of bacterial infections, and approximately 20% of cancer-inducing inflammation cases have microbial organisms as the causative agent [7]. The antibiotics used to treat diseases caused by bacteria are β-lactams, such as penicillins, cephalosporins, monobactams, carbapenems, glycopeptides, aminoglycosides, and quinolones [8]. The use of NSAIDs (diclofenac, indomethacin) is a staple in medicine, although overuse can lead to undesirable side effects [9]. There is a variety of medicinal plants in Mexico that are used for the treatment of these conditions [10,11]. 

A wide variety of medicinal applications have been attributed to species of the genus *Salvia*. This genus is the most diverse of the family Lamiaceae, with approximately 1000 species distributed in different areas of the planet [12]. Central and South America alone have 500 species. *Salvia* is the second most diverse genus in Mexico, where 300 species are endemic to the country [13,14]. Its greatest distribution occurs in mountainous areas [15,16], such as the Occidental Sierra Madre, Eastern Sierra Madre, and the highlands of Chiapas [17]. It has been reported that some species of this genus possess properties such as antimicrobial, spasmolytic, central nervous system depressants, antitumor, and anti-inflammatory [18,19,20].

*Salvia lavanduloides* Kunth, an endemic plant of Mexico, is distributed around 21 states of the Republic, from Sinaloa and Durango to Oaxaca, Veracruz, and Chiapas. In Mexico, *S. lavanduloides* has several common names, such as lucema, mazorquilla, alucema, aguanta-tsitsiki, maroon chia, flor de cielo, and k’uironi simarroni [21]. This plant is used in Mexican traditional medicine to wash wounds and as an antipyretic. It has also been used to help with gastrointestinal issues, such as stomach and gallbladder ailments, toothaches, diarrhea, and vomiting. The leaves have also been used to treat gynecological conditions, such as menstruation, hemorrhaging, and abnormal vaginal discharges [22,23,24,25,26]. As ethnomedical use indicates, some of the problems for which the species is used are associated with bacterial infections and inflammation. 

From this plant species, salviandulins **A**, **B**, and **C** (a group of secoclerodane-type diterpenoids) have been isolated and characterized [27,28]; however, there is only one pharmacological report evaluating eupatorin in the model of phorbel ester-induced pinna edema (TPA) [29]. The objective of the present work was to evaluate the antimicrobial and anti-inflammatory effect of extracts, fractions, and isolated compounds from the leaves of *Salvia lavanduloides* using the MIC method and phorbol ester-induced edema (TPA). 

## 2. Results

### 2.1. Structural Elucidation of Compound *(**1**)*, Salviandulin A *(**2**)*, and Eupatorin *(**3**)*

Compound **1** (10 mg) was isolated as a white powder, soluble in methanol. The ^13^C NMR spectrum showed twenty signals, and DEPT analysis presented eight quaternary, six methines, five methylenes, and one methyl, indicating a diterpene. Based on spectroscopic data analysis (Table 1, see Appendix A) and the comparison with data described in the literature [30] this compound corresponds to a diterpene of type neo-clerodane known as 15,16-epoxy-10-β-hydroxy-neo-cleroda-3,7,13(16), 14-tetraene-17,12R:18,19-diolide (**1**). Figure 1.

Compound **2** (73.9 mg) was isolated as a white powder, and it is soluble in methanol. The ^13^C NMR spectrum showed twenty signals and two additional ones from an acetate group (CH_3_COO). According to the DEPT experiment, seven are quaternary, eight methines, four methylenes, and one methyl. Based on spectroscopic data analysis (Table 1, see Appendix A) and the comparison with data described in the literature [27], this compound corresponds to a diterpene known as salviandulin A (**2**). Figure 1.

Compound (**3**) was isolated as a yellow powder. ^1^H NMR spectra showed three ring systems: system A [*δ* 6.86 (1H, s, H-8)]; aromatic ABX system [*δ* 7.48 (1H, d, 1.4 Hz, H-2′), 7.55 (1H, dd, 1.4, 8.4 Hz, H-5′) and 7.10 (1H, d, 8.4 Hz, H-6′)]; and a double bond system in *δ* 6.86 (1H, s, H-3), indicating a flavone nucleus called 6-hydroxyluteolin. Additionally, three proton signals of oxygenated base were observed in *δ* 3.79, 3.92, and 3.97, corresponding to methoxy groups. Based on this information (see Appendix A) the natural product was identified as 3′,5-dihydroxy-4′,6,7-trimethoxyflavone, or eupatorin (**3**), and was also consistent with spectroscopic data (see Table 2) described in the literature [31]. Figure 1.

### 2.2. Antimicrobial Activity of Extract, Fractions, and Compounds of S. lavanduloides

Table 3 shows the observed antimicrobial activity from the extracts against the strains evaluated. It is worth noting that, while Sl-AcOEt and Sl-D both showed antimicrobial activity against all strains tested, Sl-D presented the best activity with an MIC for Sa, SaR, Pa at 100, 100, and <25 μg/mL, respectively. 

Table 4 shows the antibacterial activity shown by the fractions and compounds against the strains evaluated. The antimicrobial activity of the SlD-2 fraction shows that the inhibition of Sa, Se-2, Se-3, and Sh occurred at a concentration of 100 µg/mL, while for SaR and Se-1, it was 50 µg/mL. The antimicrobial activity in Ef, Pa, and Sd occurred at MIC < 25 µg/mL; however, in the case of Ca, the inhibition occurred at >200 µg/mL. For the SlD-3 fraction, the inhibition in Sa, SaR, Se-2, Se-3 was 200 µg/mL. For Sd and Ca, they presented an inhibition >200 µg/mL, highlighting that the MIC in Se-1, Ef, and Pa was <25 µg/mL. It should be noted that the antimicrobial activity of the SlD-2 and SlD-3 fractions in Ef and Pa was maintained with a MIC < 25 µg/mL, as in the Sl-D extract. However, when the isolated compounds were evaluated, the antimicrobial activity increased, obtaining those compounds **1**–**3** presenting an inhibition of 16 µg/mL for Sa, SaR, Se-1, Se-2, Se-3, Sh, Sd, only in compounds **2**, **3** in Sd and Ca. It should be noted that the MIC < 2 µg/mL was obtained in Ef and Pa for the tree compounds, while in Sd, this concentration was obtained in compounds **1**–**3**. The MIC < 2 µg/mL was in Ca when evaluating compound **1**. The evaluation of the isolated compounds maintained the antimicrobial activity of the Sl-D extract.

### 2.3. Anti-Inflammatory Activity of Extracts, Fractions, and Compounds of S. lavanduloides

The local application of TPA in the mouse’s ear caused an edema of 11.76 mg, followed by administration of the non-steroidal anti-inflammatory drug (NSAID) indomethacin (INDO) to counteract it. Edema weight was reduced to 2.72 mg, representing an inhibition percentage of 76.87%. Table 5 presents the data of the anti-inflammatory activity of extracts, fractions, and compounds obtained from *S. lavanduloides*. The least polar extract (Sl-Hex) exhibited the highest anti-inflammatory activity, while Sl-D and Sl-AcOEt had a similar level of effect. The chemical separation was performer based on the results obtained from the antimicrobial activity; chemical separation of the dichloromethane extract was conducted. Two SlD-2 and SlD-3 fractions were generated, of which SlD-3 showed an anti-inflammatory effect. Continuing with the chemical separation of the most active fraction, SlD-3, the isolation of compounds **1**, **2**, and **3** was achieved. Mouse pinna edema was decreased with compounds **2** and **3**, both showing similar activity when compared to the positive control indomethacin (Table 5).

## 3. Discussion

The interest in the phytochemical and pharmacological study of species of the genus *Salvia* has been provoked by its wide range of traditional medicinal uses in Mexico [32]. However, there have been no previous pharmacological references demonstrating the antimicrobial and anti-inflammatory effect of *S. lavanduloides*. In this work, it was demonstrated that all evaluated organic extracts of *S. lavanduloides* inhibited TPA-induced mouse ear edema. Dichloromethane extract (Sl-D) showed the highest antimicrobial activity, as well as significant anti-inflammatory activity. The phytochemical analysis of this extract allowed the isolation of the diterpene secoclerodane salviandulin A (**2**) and the flavone eupatorin (**3**). It is important to mention that both metabolites were previously identified in the genus *Salvia* [27,29,33]. 

In the TPA test, it was discovered that eupatorin (**3**) was the most active compound, followed by salviandulin A (**2**). This drug is widely used as a reference standard. The mechanism by which it exerts its anti-inflammatory effect is by inhibiting the enzyme cyclooxygenase (COX-1) [30]. The use of NSAIDs is a staple in medicine, although overuse can lead to undesirable side effects, such as gastric ulcers. Therefore, much of the medicinal plant research focuses on finding new anti-inflammatory treatments. For example, the gender *Salvia* has been used to treat inflammation, but only few pharmacological studies have been reported with this species [18,19,20].

These results are consistent with those previously reported [34]. Our experiment used the same model cited in the literature, except with a concentration of 1 mg/ear [29]. Comparing the percentages of inhibition of anti-inflammatory activity obtained, eupatorin (**3**) performed better in our experiment, with 72.45% inhibition of atrial edema versus 56.40% inhibition previously reported [29].

In recent decades, there has been an increase in the study of biological activities of diterpenes and flavones, such as salviandulin A (**2**) and eupatorin (**3**), which have been shown to have anti-inflammatory properties. In the case of eupatorin (**3**), inhibition of iNOS, COX-2, and tumor necrosis factor alpha (TNF-α) has been reported; in addition, it can also produce NO. It also inhibited carrageenan-induced swelling of mouse leg edema [35]. In the case of salviandulin A (**2**), none of its biological activities have been reported. Therefore, we are for the first time reporting that salviandulin A (**2**) exhibits anti-inflammatory properties. However, additional studies aimed at understanding the mechanisms of action of both salviandulin A (**2**) and eupatorin (**3**) are needed. In the literature, the isolation and structural elucidation of diterpenoid-type compounds, such as salviandulin E and D [27,28,31], have been reported; however, the metabolite reported here as salviandulin A has not been reported to date, and with it, no pharmacological activity. The findings documented herein support the use of *S. lavanduloides* in traditional medicine due to the isolated diterpenes and flavonoids presented in this plant, since antimicrobial and anti-inflammatory activity was proven.

## 4. Materials and Methods

### 4.1. Equipment and Reagents

NMR spectra were recorded on an Agilent DD2-600 at 600 MHz for ^1^H and 150 MHz for ^13^C NMR, using CDCl_3_ (compounds **1** and **2**), CD_3_COCD_3_ (compound **3**) as the solvent. Bidimensional experiments (COSY, HSQC, and HMBC). Chemical shifts are reported in ppm relative to TMS. 

### 4.2. Plant Material

*Salvia lavanduloides* Kunth (10 plants, 1–1.5 m tall) leaves (all growth stages) were collected in San Juan Tlacotenco, Tepoztlán Morelos, in February 2020, and a specimen was deposited in the herbarium of the Center for Research in Biodiversity and Conservation (CIByC) of the Autonomous University of the State of Morelos (HUMO) with registration number: (HUMO-39806). 

### 4.3. Extracts

The fresh leaves of *S. lavanduloides* (1 Kg) were dried at room temperature and stored away from light. The dry material (100 g) was crushed and macerated for 24 h sequentially (in triplicate) with two liters of ascending polarity solvents, i.e., n-hexane, ethyl acetate, and dichloromethane (Merck, Darmstadt, Germany). Each extract was filtered and concentrated under reduced pressure using a rota-evaporator (Heidolph at 50 °C) to obtain extracts Sl-Hex, Sl-AcOEt, and Sl-D, respectively. 

### 4.4. Isolation and Identification of Compounds *(**1**–**3**)*

Sl-D extract (4.3 g) was adsorbed into 5 g of silica gel (silica gel 60, Merck, Darmstadt, Germany) and added to a glass column packed with silica gel (30 g, 70–230 mesh, Merck, Darmstadt, Germany). The circumvention system consisted of a gradient of hexane-ethyl acetate with polarity increments of 10%, collecting 36 fractions of 100 mL each. The fractions were concentrated at reduced pressure and monitored with normal phase chromatography and revealed with cerium sulfate. Fractions showing similarity in their chemical characteristics were grouped into 4 fractions (SlD-1 to SlD-4).

The SlD-3 fraction (584 mg) was then adsorbed in silica gel (RP-18, Merck, Darmstadt, Germany) for separation using a pre-packed reverse phase column (10 g, RP-18, Merck, Darmstadt, Germany) as a stationary phase. The elution system consisted of 100% acetonitrile (HPLC grade, Merck, Darmstadt, Germany) and polarity increments of 10% water, collecting volumes of 7 mL. Separation was monitored with reverse phase thin layer chromatography (RP-18, Merck, Darmstadt, Germany) and revealed with cerium sulfate. In fraction 21–22, the diterpene 15,16-epoxy-10-β-hydroxy-neo-cleroda-3,7,13(16), 14-tetraene-17,12R:18,19-diolide (**1**, 10 mg) was isolated. Fractions 26–28 had salviandulin A (**2**, 25 mg), and in fractions 37 and 38, a flavone called eupatorin (**3**, 85.2 mg) was identified (Figure 2).

### 4.5. Antimicrobial Activity

#### 4.5.1. Bacterial

Extracts, fractions, and compounds were evaluated against Gram-positive bacteria: *Staphylococcus aureus* ATCC 29213 (Sa) and methicillin-resistant *Staphylococcus aureus* ATCC 42310 (SaR), *Staphylococcus epidermis* ATCC 35984 (Se-1), *Staphylococcus epidermis* ATCC 12228 (Se-2), *Staphylococcus epidermis* ATCC 1042 (Se-3), *Staphylococcus haemolitycus* ATCC 1165 (Sh), *Enterococcus faecalis* ATCC 29212 (Ef); and Gram-negative bacteria *Pseudomonas aeruginosa* ATCC 27853 (Pa), *Salmonella dublin* NTCC 9676 (Sd). In addition, all samples were also evaluated against the fungus *Candida albicans* ATCC 10231 (Ca).

#### 4.5.2. Minimal Inhibitory Concentration (MIC)

Following a methodology previously described [36], all bacteria were standardized to Mc Farland’s 0.5 (1.5 × 10^8^ mL/mL CFUs). The samples were diluted with dimethyl sulfoxide (DMSO, 200 μL), water (800 μL), and evaluated at concentrations of 200, 100, 50, and 25 μg/mL for extract and fractions, and pure compounds 16, 8, 4, and 2 μg/mL in microplates of 96 flat-bottomed wells, where 100 μL of Mueller Hinton broth (MHB, Merck, Darmstadt, Germany) and serial dilutions of the extracts were placed. An amount of 2 μL of bacterial strains were inoculated and incubated at 37 °C for 24 h. Gentamicin (10 μg/mL) was used as a positive control and DMSO as the negative control. All experiments were conducted in triplicate [36].

### 4.6. Anti-Inflammatory Activity

#### 4.6.1. Animals

Male Balb-C mice (20 g) were obtained from Centro Médico Nacional SXXI-IMSS. They were organized in groups of 7 animals under controlled conditions with a 12 h light–dark cycle, at 20 ± 1 °C of temperature and free access to food (Labdiet 5008, Brentwood, MO, USA) and water. Mice were handled according to the Mexican Official Standard on the Care and Management of Animals (NOM-062-ZOO-1999) and international regulations. The Research Committee of IMSS approved the experimental designs used in the present study F-2021-1702-022.

#### 4.6.2. 12-O-tetradecanoylphorbol-13-acetate (TPA)-induced Mouse Ear Edema

Animal inflammation was induced following a method previously described [37,38]. Eight groups of seven mice each were formed. Then, TPA (Sigma Chemical Co. St. Louis, MO, USA, 2.5 g) dissolved in acetone (Baker, 20 L) was applied in the internal and external surface on the right ear to cause edema. Doses of 1 mg/ear of each treatment (Sl-Hex, Sl-AcOEt, Sl-D, SlD-2, SlD-3, salviandulin-A, eupatorin) were applied on the ear of each individual. Reference anti-inflammatory drug (indomethacin, Sigma Aldrich, St. Louis, MI, USA) was administered at 1 mg/ear. All treatments were dissolved in acetone and applied topically on ears immediately after the administration of TPA. Six hours after administration of the inflammatory agent, the animals were sacrificed by cervical dislocation. Circular sections of 6 mm in diameter were taken from both the treated (t) and the non-treated (nt) ears, which were weighed to determine the inflammation. The percentage of inhibition was obtained using the expression below:Inhibition % = [(Δw control − Δw treatment/ΔW)] × 100(1)
where Δw = wt − wnt; wt is the weight of the section of the treated ear; wnt is the weight of the section of the non-treated ear.

### 4.7. Statistical Analysis

Statistical evaluation was performed with SPSS (Version 22.0) software based on one-way ANOVA with a confidence level of 95%, * *p* ≤ 0.05, and followed by Dunnett’s test with one-tale: * *p* ≤0.05 in comparison with the negative control.

## 5. Conclusions

In this study, the anti-inflammatory and antimicrobial activity of the species *Salvia lavanduloides* Kunth was demonstrated by testing the three isolated compounds (diterpenoid (**1**), salviandulin A (**2**), and eupatorin (**3**)). Our data revealed that compound **1** was the most bioactive against all the tested microorganisms. It was also found that both salviandulin A and eupatorin present high anti-inflammatory activity at 70% and 72%, respectively, for mouse pinna edema. Furthermore, the elucidation of the chemical structures of the three isolated and characterized natural products (**1** to **3**) highlights the utility of the protocol utilized and opens up an opportunity to foster drug discovery, while validating the use of this plant in traditional medicine. 

## Figures and Tables

**Figure 1 plants-11-01739-f001:**
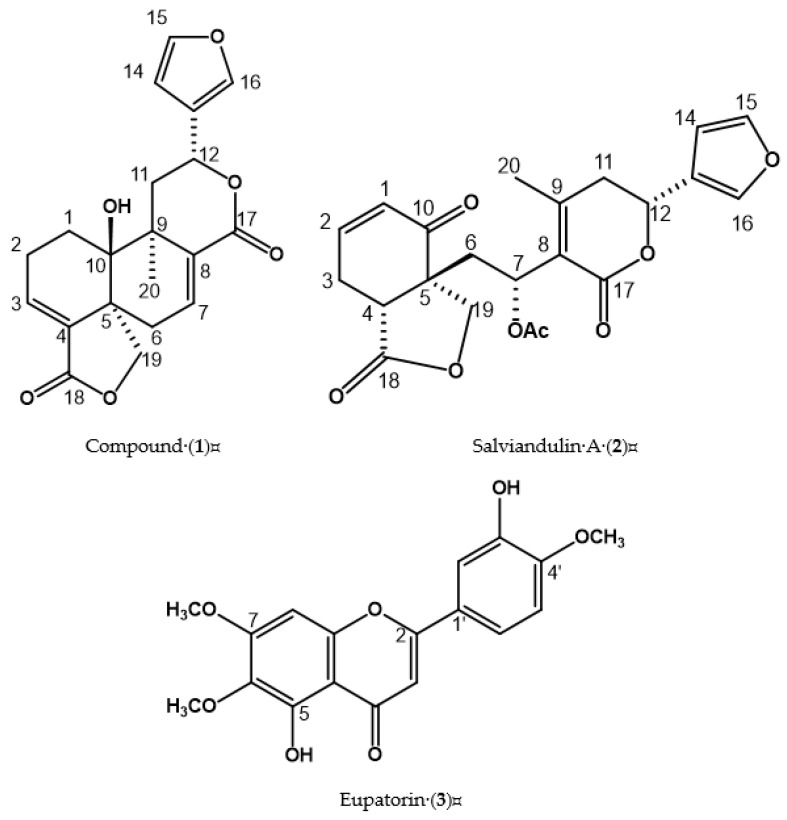
Chemical structures of compounds isolated from the SlD-3 fraction.

**Figure 2 plants-11-01739-f002:**
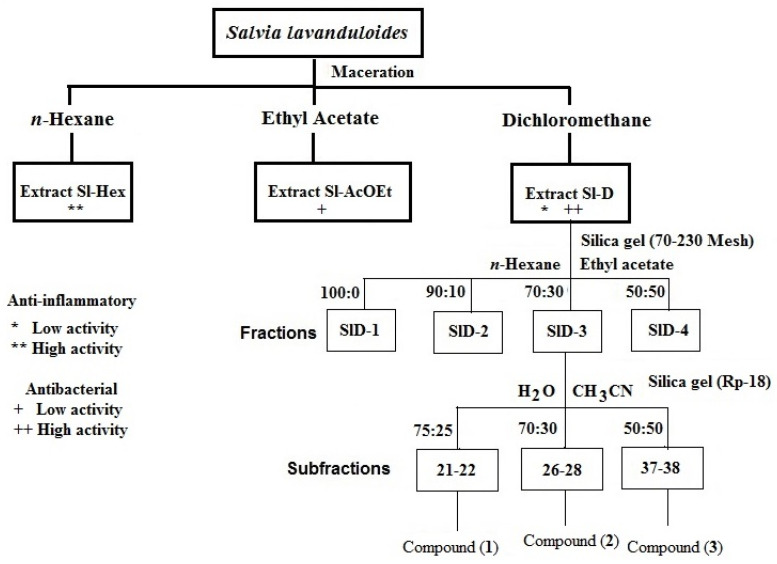
Protocol used to obtain all extracts, fractions, and compounds of *Salvia lavanduloides*. 15,16-epoxy-10-β-hydroxy-neo-cleroda-3,7,13(16),14-tetraene-17,12R:18,19-diolide (**1**), salviandulin A and (**2**) eupatorin (**3**).

**Table 1 plants-11-01739-t001:** NMR spectroscopic data for compounds **1** and **2** (600 MHz, CDCl_3_, δ ppm).

Position	*δ*^1^H (J in Hz) 1	*δ*^13^C 1	*δ*^1^H (J in Hz) 2	*δ*^13^C 2
1a b	1.79, dd(8.0, 14.3) 1.72, ddd(4.0, 11.3, 15.4)	25.3	6.13, ddd(1.1, 2.5, 10.2)	129.7
2a b	2.40, dd(4.0, 8.0) 2.41, dd(4.0, 8.0)	22.9	6.98, dddd(0.7, 2.5, 5.5, 10.2)	149.3
3a b	6.79, dd(3.3, 3.6)	135.6	3.01, dd(br, 5.1, 20.1) 2.76, ddd(2.5, 6.9, 20.5)	23.2
4		131.84	3.07, d(br, 6.9)	43.3
5		44.6		51.6
6a b	2.79, dd(4.4, 20.5) 2.33, d(1.8, 20.5)	39.10	2.69, dd(8.0, 14.6) 2.33, d(4.4, 14.6)	39.1
7	6.42, dd(3.3, 4.0)	133.1	5.63, d(4.4, 8.0)	67.3
8		134.8		124.9
9		40.82		154.8
10		71.7		197.4
11a b	2.66, dd (2.2, 15.4) 1.70, dd(12.4, 15.4)	38.5	2.74, m 2.51, dd(3.6, 17.9)	37.5
12	5.03, dd(1.83, 12.4)	71.91	5.3, d(3.6, 11)	71.0
13		123.7		123.5
14	6.63 d(br, 1.4)	109.03	6.42, d(br, 1.1)	108.5
15	7.69, dd(1.4, 1.8)	144.0	7.42, dd(1.8, 1.8)	143.9
16	7.79, d(br, 0.7)	140.58	7.47, dd(0.7, 0.7)	140.1
17		167.8		163.0
18		168.96		176.5
19a b	4.29, d(8.0) 4.33, d(8.0)	73.35	4.74, d(8.8) 3.84, d(8.8)	73.6
20	1.10, s	29.20	2.13, s	20.4
MeCO_2_-			2.03, s	170.2, 21.1

**Table 2 plants-11-01739-t002:** NMR spectroscopic data for eupatorin (**3**, 600 MHz, CD_3_COCD_3_, δ ppm, *J* in Hz).

Position	δ^1^H (in ppm, J in Hz) 3	δ ^13^C 3
2		165.3
3	6.67(s)	104.2
4		183.5
5		154.0
6		133.4
7		160.0
8	6.86(s)	92.0
9		153.5
10		106.3
1′		124.4
2′	7.48 (d, 1.4)	112.6
3′		148.0
4′		152.1
5′	7.55 (dd, 1.4, 8.4)	113.7
6′	7.10 (d, 8.4)	119.6
6-OCH_3_	3.79 (s)	56.3
7-OCH_3_	3.92(s)	56.8
4′-OCH_3_	3.97(s)	60.5

**Table 3 plants-11-01739-t003:** Antimicrobial activity of *Salvia lavanduloides* extracts.

	Extracts μg/mL				
Bacterium	Sl-Hex	Sl-AcOEt	Sl-D	Control (+)	Control (−)
Sa	>200	200	100	---	†
SaR	>200	200	100	---	†
Se-1	100	<25	100	---	†
Se-2	>200	200	200	---	†
Se-3	>200	200	200	---	†
Sh	>200	200	200	---	†
Ef	<25	<25	<25	---	†
Pa	<25	<25	<25	---	†
Sd	<25	<25	<25	---	†
Ca	<25	<25	<25	†	†

(---): No growth; (†) growth. *Staphylococcus aureus* ATCC 29213 (Sa), methicillin-resistant *Staphylococcus aureus* ATCC 42310 (SaR), *Staphylococcus epidermis* ATCC 35984 (Se-1), *Staphylococcus epidermis* ATCC 12228 (Se-2), *Staphylococcus epidermis* ATCC 1042 (Se-3), *Staphylococcus haemolitycus* ATCC 1165 (Sh), *Enterococcus faecalis* ATCC 29212 (Ef), Gram-negative bacteria *Pseudomonas aeruginosa* ATCC 27853 (Pa), *Salmonella dublin* NTCC 9676 (Sd), and the fungus *Candida albicans* ATCC 10231 (Ca). n-hexane (Sl-Hex), ethyl acetate (Sl-AcOEt), and dichloromethane (Sl-D).

**Table 4 plants-11-01739-t004:** Antimicrobial activity of *Salvia lavanduloides* fractions and compounds.

	Fractions μg/mL		μg/mL Compounds				
Bacterium	SlD-2	SlD-3	1	2	3	Control (+)	Control (−)
Sa	100	200	16	16	16	---	†
SaR	50	200	16	16	16	---	†
Se-1	50	<25	16	16	16	---	†
Se-2	100	200	16	16	16	---	†
Se-3	100	200	16	16	16	---	†
Sh	100	100	16	16	16	---	†
Ef	<25	<25	<2	<2	<2	---	†
Pa	<25	<25	<2	<2	<2	---	†
Sd	<25	>200	<2	16	<2	---	†
Ca	>200	>200	<2	16	16	†	†

(---): does not grow; (†) growth. *Staphylococcus aureus* ATCC 29213 (Sa), methicillin-resistant *Staphylococcus aureus* ATCC 42310 (SaR), *Staphylococcus epidermis* ATCC 35984 (Se-1), *Staphylococcus epidermis* ATCC 12228 (Se-2), *Staphylococcus epidermis* ATCC 1042 (Se-3), *Staphylococcus haemolitycus* ATCC 1165 (Sh), *Enterococcus faecalis* ATCC 29212 (Ef), Gram-negative bacteria *Pseudomonas aeruginosa* ATCC 27853 (Pa), *Salmonella dublin* NTCC 9676 (Sd), and the fungus *Candida albicans* ATCC 10231 (Ca). SlD-2 and SlD-3 fractions. 15,16-epoxy-10-β-hydroxy-neo-cleroda-3,7,13(16),14-tetraene-17,12R:18,19-diolide (**1**), salviandulin A (**2**), and eupatorin (**3**).

**Table 5 plants-11-01739-t005:** Anti-inflammatory activity of *Salvia lavanduloides*.

Treatments (1.0 mg/ear)	Edema (mg)	Inflammation Inhibition (%)
VEH	11.76 ± 1.40	0
INDO	2.72 ± 0.75	76.87
Extracts		
Sl-Hex	4.46 ± 2.04	62.02
Sl-D	6.50 ± 1.23	44.73
Sl-AcOEt	6.45 ± 1.10	45.15
Fractions		
SlD-2	9.73 ± 2.14	17.23
SlD-3	4.56 ± 1.40	61.22
Compounds		
Salviandulin A (2)	3.50 ± 0.67	70.24
Eupatorin (3)	3.25 ± 1.01 *	72.45

Extracts, fractions, and compounds *S. lavanduloides*, TPA (acetone); INDO, indomethacin. Data show mean ± S.E.M. of seven animals. * *p* < 0.05 indicates statistically significant differences using ANOVA followed by the Dunnett test compared with VEH-Tween group.

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
