# Peer review of "Eupatorin and Salviandulin-A, with Antimicrobial and Anti-Inflammatory Effects from Salvia lavanduloides Kunth Leaves"

_plants, 2022, doi:10.3390/plants11131739_

Round 1

Reviewer 1 Report

The manuscript plants-1774599 entitled "Eupatorin and salviandulin-A, with antimicrobial and anti-inflammatory effect of Salvia lavanduloides Kunth leaves" reported the antimicrobial and anti-inflammatory properties of two compounds extracted from endemic salvia specie.

This is great research work. It is well written; the introduction section does explain what was done on the topic analyzed. In the same way, a deep review of pertinent literature was used. The materials and methods are very clear allowing their reproduction in any laboratory. The scientific quality of the results and discussion section is so good. The results are presented in a clear and concise way. The authors discuss and compare their findings with previous reports in this field.

The only serious issue to improve will be The English language. In its current state, the level of English throughout your manuscript does not meet the journal's desired standard. There are some grammatical and spelling errors and full stops missing as well as instances of badly worded/constructed sentences. Please check the manuscript and refine the language carefully. I suggest that you should ask several colleagues who are skilled authors to check the English before your submission.

  The conclusion section is redundant with results already summarized and analyzed. In the present form same as the abstract of the paper. This section is not adequate and should be improved. Rewrite, please

Reviewer 2 Report

The manuscript entitled "Eupatorin and salviandulin-A, with antimicrobial and antiinflammatory effect of Salvia lavanduloides Kunth leaves" brings a simple, but interesting preliminary study regarding the anti-inflammatory and antimicrobial effects of extracts, fractions and isolated compounds of S. lavanduloides leaves. The paper is fixes well as a "communication", since the results presented are preliminary and more analysis should be conduct in order to have a complete research paper. However, despite being interesting, the manuscript needs several corrections before its acceptance. First of all, the authors must revise their English, since some mistakes and unclear sentences can be found. Even Spanish words are presented in the text. In addition, the references must be presented as required by the Journal's guideliness and the authors must check them. Besides, the main following corrections are needed:

- M&M: it needs several detailments. Besides, how many plants were used in the study? Which kind of leaves were collected (young, mature?) and in which season? These information must be presented. The NMR conditions and the description of this analysis are not presented. How was it conducted? 

- Results: some results are simply described. Besides, the statistical analysis are almost missing (only one sample in the table 3 presents statistical results). In the tables, the authors should check the SE values. Some are half of the mean and one is even higher. This should be carefully checked and maybe repeated. In addition, several parts in the "results" are actually "discussion" sentences. The antimicrobial results have no statistics at all. With some of the values presented is difficult to affirm if a sample is better than other.

- Discussion: it must be improved. The repetition of the results must be avoided.

All my other comments, suggestions and corrections can be found in the PDF here attached. 

Based on all above I recommend "Major Review" to the manuscript.

Round 2

Reviewer 2 Report

The authors have answered all my querries and corrected all my suggestions. The manuscript can be accepted.